# The Inflammatory and Oxidative Status of Newly Diagnosed Class III and Class IV Lupus Nephritis, with Six-Month Follow-Up

**DOI:** 10.3390/antiox12122065

**Published:** 2023-12-01

**Authors:** José Ignacio Cerrillos-Gutiérrez, Miguel Medina-Pérez, Jorge Andrade-Sierra, Alejandra De Alba-Razo, Fermín Paul Pacheco-Moisés, Ernesto Germán Cardona-Muñoz, Wendy Campos-Pérez, Erika Martínez-López, Daniela Itzel Sánchez-Lozano, Andrés García-Sánchez, Tannia Isabel Campos-Bayardo, Alejandra Guillermina Miranda-Díaz

**Affiliations:** 1Department of Nephrology, National Medical Center of the West, Mexican Social Security Institute, Guadalajara 44340, Jalisco, Mexico; chachisnefro@hotmail.com (J.I.C.-G.); enepis1@yahoo.com.mx (M.M.-P.); jorg_andrade@hotmail.com (J.A.-S.); 2Department of Physiology, University Center of Health Sciences, University of Guadalajara, Guadalajara 44360, Jalisco, Mexico; alejandra.dealba8671@alumnos.udg.mx (A.D.A.-R.); cameg1@gmail.com (E.G.C.-M.); itzel.10274@gmail.com (D.I.S.-L.); andres_garciasanchez_3@hotmail.com (A.G.-S.); tanniaisabelcb@gmail.com (T.I.C.-B.); 3Department of Chemistry, University Center of Exact Sciences and Engineering, University of Guadalajara, Guadalajara 44430, Jalisco, Mexico; ferminpacheco@hotmail.com; 4Department of Molecular Biology and Genomics, Institute of Nutrigenetics and Translational Nutrigenomics, University of Guadalajara, Guadalajara 44340, Jalisco, Mexico; wendy_yareni91@hotmail.com (W.C.-P.); erikamtz27@yahoo.com.mx (E.M.-L.)

**Keywords:** oxidative stress, lupus nephritis, antioxidants, systemic lupus erythematosus, pro-inflammatory cytokines

## Abstract

Lupus nephritis (LN) is the most frequent and severe complication of systemic lupus erythematosus (SLE). A prospective cohort with a six-month follow-up was performed. Twelve SLE patients diagnosed with LN Class III, twelve NL Class IV patients, and twelve healthy control subjects (HC) were included. SLE data, renal function, oxidants, antioxidants, and inflammation were determined at baseline and six-month follow-up. During the six-month follow-up, the SLE Disease Activity Index (SLEDAI-2K) decreased in both LN Class III (20.08 ± 6.92 vs. 11.92 ± 5.87, *p* < 0.001) and LN Class IV (25.33 ± 6.01 vs. 13.83 ± 5.52, *p* < 0.001) patients. Furthermore, the values of the C4 component also increased during follow-up for LN Class III (25.36 ± 6.34 vs. 30.91 ± 9.22, *p* = 0.027) and LN Class IV (12.18 ± 3.90 vs. 20.33 ± 8.95, *p* = 0.008) groups. Regarding inflammation markers, both groups presented decreased C-reactive protein (CRP), but this was only significant for patients with LN class III (7.93 ± 1.77 vs. 4.72 ± 3.23, *p* = 0.006). Renal function remained stable in both groups, with no changes in eGFR. Patients with LN Class III and Class IV showed higher baseline levels for lipoperoxides (Class III *p* < 0.01, Class IV *p* < 0.1) and carbonyl groups in proteins (Class III *p* < 0.01, Class IV *p* < 0.1) compared to HC. Moreover, both groups presented lower baseline values of total antioxidant capacity (Class III *p* < 0.01, Class IV *p* < 0.1) and catalase (Class III *p* < 0.01, Class IV *p* < 0.1) compared to HCs. However, antioxidant and oxidant markers did not show significant differences between baseline values and at six months for either of the two study groups. In conclusion, patients show an imbalance in the oxidative state characterized by the increase in the oxidants LPO and protein carbonyl groups and the decrease in the activity of the antioxidant enzymes TAC and CAT compared to HC. However, the patients did not present an increase in disease activity and renal function improvement. The glomerular filtration rate did not change during the length of the study, and SLEDAI -2K, C3, and C4 improved. The early co-management between Rheumatologists and Nephrologists is essential to prevent the rapid progression of LN. It would be interesting to administer antioxidant supplements to patients with a recent diagnosis of LN and evaluate its effect in a follow-up study.

## 1. Introduction

Systemic lupus erythematosus (SLE) is a chronic relapsing multisystem autoimmune disorder that primarily affects women of reproductive age. The estimated incidence is 1.25 per 100,000 persons in the United States and Europe [1]. SLE is more frequent among non-white populations, with higher reported prevalence among Africans and Caribbeans [2]. SLE is characterized by the presence of auto-reactive T and B lymphocytes and the production of autoantibodies against nuclear and cytoplasmic antigens [3]. SLE is associated with chronic inflammation characterized by a complex pathogenesis not fully known and understood [4]. Between 50 and 80% of SLE patients will develop LN over the course of the disease [4].

LN constitutes a poor prognostic feature because of its ability to increase morbidity and mortality in SLE patients [5]. The consensus was derived from the 18-member meeting of an international nephropathology working group in Leiden, The Netherlands, in 2016 and modified in 2017 [6]. Considerable studies in patients and animal models have been reported implicating the role of oxidative stress (OS) in the pathogenesis of SLE [7]. The production of free radicals (FR) is an integral part of normal metabolism. If normal metabolism is uncontrolled, OS will occur. OS damage to lipids, proteins, nucleic acids, and carbohydrates is detrimental and concomitant with SLE. OS biomarkers can determine the extent of oxidative injury and indicate the source of oxidation [8]. Lipids are susceptible targets of oxidation, and lipid peroxidation products are potential biomarkers for establishing the oxidative status of disease [9,10]. Lipid peroxidation generates a variety of relatively stable end products of decomposition, mainly unsaturated reactive aldehydes, such as malondialdehyde (MDA), hexanoyl-lysine adducts (HEL), HNE, and 2-propenal (acrolein) and others which can be measured as an indirect index of OS [11,12]. Carbonyl groups in protein are formed by direct oxidation of specific amino acid residues, particularly lysine, arginine, threonine, proline, and histidine, or by secondary reaction with products of lipid peroxidation or glycoxidation reaction with the lysine group [13]. To counteract the damaging effect of oxidants, the body designed antioxidant enzymes. Among the antioxidants are the enzyme superoxide dismutase (SOD), catalase (CAT), glutathione peroxidase (GPx), total antioxidant capacity (TAC), in addition to various vitamins such as ascorbic acid, β-carotene, Zn, selenium, Cu, Fe, etc. [14].

This study aimed to determine the inflammatory and oxidative status in newly diagnosed LN Class III and Class IV with a six-month follow-up.

## 2. Materials and Methods

A prospective cohort study with a six-month follow-up was performed. Women and men over 18 years old with SLE and a recent diagnosis of focal LN (Class III) or diffuse LN (Class IV) were included. The patients were selected from the Department of Nephrology of the Specialty Hospital of the National Medical Center of the West of the Mexican Institute of Social Security (IMSS) in Guadalajara, Jalisco, Mexico. The included patients agreed to participate in the study. All patients maintained their SLE treatment, which consisted primarily of glucocorticoids. Patients who ingested antioxidants (vitamin E, vitamin C, etc.) three months before the study or patients with clinical or biochemical data of an infectious process were not included. Patients with any type of diabetes or with a history of thrombotic events or polycystic diseases were also excluded. In addition, patients who withdrew informed consent were excluded from the study. Below is the LN project flowchart (Figure 1).

### 2.1. Data Collection

Gender, age, height, and body weight were recorded. Biochemical data were determined, including hemoglobin, hematocrit, platelets, glucose, albumin, chloride, potassium, phosphorus, calcium, sodium, and magnesium. Renal function data were included, including glomerular filtration rate, urea, creatinine, proteinuria, albuminuria, urinary creatinine, and hematuria. Data corresponding to SLE activity, including SLE Disease Activity Index (SLEDAI-2K) [15], C3, C4, and anti-DNA antibodies, were determined. All results were collected at the first appointment with the Nephrologist and at the six-month follow-up. Patients with LN Class III and Class IV were included according to classification by biopsy-proven diagnosis within three months after detection of LN, according to WHO or International Society of Nephrology and Society for Renal Pathology (ISN)/RPS) 2003 criteria [16]. Patients with LN Class III (focal) are characterized by proteinuria, hematuria, increased serum creatinine, sometimes nephrotic syndrome, and hypertension. The progression of the lesion depends on the percentage of affected glomeruli. LN Class IV (diffuse) patients are characterized by hematuria, proteinuria, nephrotic syndrome, hypertension, increased anti-DNA antibodies, and hypocomplementemia. These patients progress to renal failure [6].

Inflammatory status was determined by C-reactive protein (high-sensitivity CRP) and interleukin-6 (IL-6) levels. Lipoperoxides (LPO), nitrites/nitrates, and carbonyl groups in proteins were measured for oxidative status. Antioxidant status was determined by TAC, CAT, and SOD. For the previous analyses, the SynergyTM HT multi-detection microplate reader (Bio-Tek, Winooski, VT, USA) was used.

Venous blood samples (5 mL in a dry tube and 5 mL in a tube with 7.2 mg ethylenediaminetetraacetic acid di potassium (K2 EDTA)) were collected at the first visit to the Nephrology Department and the six-month follow-up. Samples were centrifuged at 3500 rpm for 10 min to obtain serum and plasma. Aliquots were stored at −80 °C for later analysis. Ten mL of extra blood was obtained from twelve blood donors who met the requirements to be blood donors. These subjects formed the healthy control (HC) group. The HC samples were used to establish the normal value of the reagents to determine the inflammatory and oxidative status in LN.

### 2.2. Oxidative Stress Markers

#### 2.2.1. LPO

The levels of LPO in plasma were measured through the FR22 assay kit (Oxford Biomedical Research Inc., Oxford, MI, USA^®^) as per the manufacturer’s instructions. The limit of detection for this test was 0.1 nmol/mL; the chromogenic reagent reacts with MDA and 4-hydroxy-alkenes to form a stable chromophore. A total of 200 μL of plasma with 650 μL of N-methyl-2-phenylindole in acetonitrile (Reagent 1) was diluted with ferric iron in methanol. Samples were agitated, after which 150 μL 37% HCl was added, followed by incubation at 45 °C for 60 min and centrifugation at 12,791 rpm for 10 min. Next, 150 μL of the supernatant was taken, and absorbance was measured at 586 nm. The pattern curve with known concentrations of 1,1,3,3-Tetramethoxypropane in Tris-HCl was used. The intra-assay CV was 8.5%. The levels are reported in Mm.

#### 2.2.2. Nitrites/Nitrates

For the determination of nitrites/nitrates, the colorimetric method with 20 µL of serum to which was sequentially administered 100 µL of 0.8% (*w*/*v*) VCl3 in 1 M H_3_PO_4_, 50 µL of 2% (*w*/*v*) sulfanilamide in 5% (*v*/*v*) H_3_PO_4_, and 50 µL of 0.2% (*w*/*v*) N-(1-naphthyl) ethylenediamine dichlorohydrate. Finally, the plate was read at 540 nm in a spectrophotometer within the first 20 min of completion of the procedure according to Tenorio, L.F.A. et al. [17].

#### 2.2.3. Carbonyl Groups in Proteins

A total of 200 μL of plasma was mixed with 500 μL of 10 mM 2, 4-dinitrophenylhydrazine in 2 M HCl and incubated for one hour at room temperature. After that, 333 μL of trichloroacetic acid (30%, *p*/*v*) was added, followed by centrifugation at 14,000× *g* for 20 min. The precipitate was washed thrice with 1 mL of ethanol-ethyl acetate solution (1:1, *v*:*v*). The final precipitate was added 600 μL of guanidine hydrochloride 6 M, followed by incubation for 15 min at room temperature. The plate was read at 370 nm according to Lenz, A.G. et al. [18].

### 2.3. Inflammation Markers

#### 2.3.1. CRP

VITROS CRP^®^ (Ortho-Clinical Diag-nostics, Inc., Rochester, NY, USA) manufacturer’s recommendations were followed using an enzymatic heterogeneous sandwich immunoassay, where a phosphorylcholine (PC) derivative is covalently bound to polystyrene polymer beads, the presence of calcium serves as the capture agent for the anti-CRP conjugated monoclonal antibody, and horseradish peroxidase (HRP) serves as the signal generator. The reflection density of the dye is measured after the addition of the Vitros immuno-wash reagent at the end of the incubation. The reflection density is directly proportional to the CRP concentration of the sample. The detection dye is washed and read at 540 nm immediately after incubation.

#### 2.3.2. IL-6

The method used is from the commercial kit 900-K25 and 900-K16 of the manufacturer Peproetch^®^ (Rocky Hill, NJ, USA) using the sandwich ELISA assay. The 96-well microplate was coated with 100 µL of the capture antibody for IL-6, followed by overnight incubation at room temperature. Subsequently, the spaces where the antibody did not bind were blocked with 300 µL of buffer for 1 h at room temperature. After four washes with buffer, 100 µL of the sample was added to the microplate and incubated for 2 h at room temperature. The plate was washed 4 times, and 100 µL of a detection antibody was added and incubated for 2 h at room temperature. After 4 washes with buffer, 100 µL of avidin-HRP conjugate was added to each well and incubated for 30 min at room temperature, then 100 µL of the substrate was added to develop color change. The absorbance of each well was recorded at 405 nm with 650 nm correction. The IL-6 standard curve was prepared to interpolate the IL-6 concentrations of each sample.

### 2.4. Antioxidants

#### 2.4.1. SOD

We followed the kit manufacturer’s instructions (SOD No. 706002, Cayman Chemical Company^®^, Ann Arbor, MI, USA) for the detection of O_2_^−^ generated by the xanthine oxidase and hypoxanthine enzymes through the reaction of tetrazolium salts. The serum samples were 5 µL in the sample buffer, 200 μL of the radicals’ detector (1:400 dilution), and 10 μL of the sample added. After slow agitation, 20 μL of xanthine oxidase was added to the wells. The microplate was incubated for 20 min at room temperature. The absorbency was read at 440 wavelengths of nm. The levels are reported in U/mL.

#### 2.4.2. Total Antioxidant Capacity

CuCl2 solution, 1.0 × 10^−2^ M, was prepared by dissolving 0.4262 g CuCl_2_. 2H_2_O in water, which was diluted to 250 mL. Ammonium acetate buffer at pH- 7.0, 1.0 M, was prepared by dissolving 19.27 g NH4Ac in water and diluting it to 250 mL. Neocuproine (Nc) solution, 7.5 × 10^−3^ M, was prepared daily by dissolving 0.039 g Nc in 96% ethanol and diluting it to 25 mL with ethanol. Trolox, 1.0 × 10^−3^ M, was prepared in 96% ethanol. The chromogenic radical reagent ABTS, at 7.0 mM concentration, was prepared by dissolving 0.1920 g of the compound in water and diluting it to 50 mL. To this solution, 0.0331 g of K_2_S_2_O_8_ was added such that the final persulfate concentration in the mixture was 2.45 mM. The resulting ABTS radical cation solution was left to mature at room temperature in the dark for 12–16 h and then used for TEAC assays according to Yildiz, L. et al. [19]. The test was carried out with 100 µL of sample, 0.33 mL of ammonium acetate solution (4 °C), 0.33 mL of copper solution, and 0.33 mL of NC solution. The absorbency was read at micromol trolox equivalents.

#### 2.4.3. Catalase

A total of 40 µL of plasma was added to a microtube with 300 µL of distilled water. Subsequently, 500 µL of hydrogen peroxide solution in sodium–potassium phosphate buffer, pH 7.4, was added. The reaction was followed for 3 min and stopped with 100 µL of ammonium molybdate solution. Then, 20 µL was added to a microplate with 180 µL of water, and the absorbance of the samples was measured at 379 nm. The activity was reported as KU/mL according to Hadwan, M.H.; Abed, H.N [20].

### 2.5. Statistical Analysis

Normally distributed data were presented as mean ± standard deviation (SD). Categorical variables were expressed as frequency and percentage. According to the type of data distribution, all demographic and related characteristics were compared between baseline and six-month follow-up determination using Chi^2^, paired-sampled *t*-test, and independent sample *t*-test. Statistical analysis was performed using IBM SPSS v.18 software (Chicago, IL, USA). Any value of *p* ≤ 0.05 was considered significant.

### 2.6. Ethical Considerations

This study was conducted by the ethical principles for medical research on human subjects stipulated in the Declaration of Helsinki 64th General Assembly, Fortaleza Brazil, October 2013, and the Standards of Good Clinical Practice according to the guidelines of the International Conference on Harmonization. To safeguard the confidentiality of patient data, codes were assigned to the samples and by the provisions of the General Health Law of Mexico, according to the Regulations of the General Health Law on Research for Health, art. 17, which corresponds to a category II study. All patients signed the informed consent form in the presence of witnesses. This study was approved by the local Ethics and Research Committees (IMSS) on 12 January 2022, with the registration number R-2022-1301-010.

## 3. Results

Twenty-four patients were included; twelve corresponded to LN Class III, twelve to LN Class IV, and twelve to HC. All patients were included in the first visit to the Nephrologist referred by the Rheumatologist for management by both specialists.

The female gender predominated with 83.3% in LN Class III and 91.7% in LN Class IV. The age of LN Class III patients was 27.83 ± 7.06 years, LN Class IV was 33.00 ± 7.87 years, and HC was 26.50 ± 3.85 years. Body weight in Kg was similar in LN Class III and LN Class IV. LN Class III patients were taller, 163.42 ± 10.80 cm, while Class IV patients were 155.92 ± 4.46 cm (*p* = 0.04) Table 1.

The SLEDAI -2K at baseline in LN Class III was 20.08 ± 6.92, and in Class IV, it was 25.33 ± 6.01, with no significant difference between both. Complement C3 fraction was significantly decreased at baseline in LN Class IV patients, 62.82 ± 11.40 mg/dL (*p* < 0.001), compared to Class III, 115.97 ± 37.80 mg/dL. The baseline determination of the complement C4 fraction showed the same behavior. Complement C4 fraction levels were higher in LN Class III patients, 25.36 ± 6.34 mg/dL, than in Class IV, 12.18 ± 3.90 mg/dL (*p* < 0.001). Basal anti-DNA levels were found to significantly increase in LN Class IV patients (73.69 ± 74.69 mg/dL) vs. Class III (25.12 ± 32.75 mg/dL) (*p* = 0.05). The biochemical data of renal function were similar at baseline between NL Class III and Class IV patients Table 1.

Table 2 presents the baseline vs. six-month follow-up results for LN Class III and Class IV patients. It is observed that the SLEDAI-2K decreased significantly between the LN Class III baseline result, 20.08 ± 6.92 -2K versus 11.92 ± 5.87 -2K at six-month follow-up (*p* < 0.001). The same behavior was observed in the outcome at the six-month follow-up in NL Class IV, 13.83 ± 5.52 versus baseline, 25.33 ± 6.01 (*p* < 0.001). Hematocrit improved in LN Class IV at six-month follow-up with 39.29 ± 6.75% vs. 35.76 ± 5.41% (*p* = 0.009). Glucose levels decreased significantly at the six-month follow-up in LN Class IV, 82.45 ± 9.82 mg/dL vs. baseline at 98.77 ± 12.34 mg/dL (*p* = 0.005). A significant decrease in chloride levels was found at a six-month follow-up for Class III (*p* = 0.007) and Class IV (*p* = 0.05). Urea levels decreased in LN Class IV at the six-month follow-up with 33.93 ± 10.46 mg/dL vs. baseline levels of 43.90 ± 18.35 (*p* = 0.012). Proteinuria decreased significantly between baseline in LN Class III, 2.16 ± 1.62 g/L in contrast at six-month follow-up, 1.26 ± 1.13 g/L (*p* = 0.02). In LN Class IV, proteinuria decreased at six-month follow-up, 2.33 ± 1.89 g/L (*p* = 0.002) versus 3.33 ± 1.98 g/L at baseline. Urinary creatinine significantly increased at the six-month follow-up in LN Class IV, 1.69 ± 0.41 mg/dL vs. baseline 1.36 ± 0.30 mg/dL (*p* = 0.048). Hematuria was significantly decreased at the six-month follow-up in Class III, 27.00 ± 56.41 g/L versus baseline determination, 42.25 ± 69.89 g/L (*p* = 0.012).

Basal determination of CRP obtained similar levels between HC, LN Class III, and Class IV. Significantly decreased levels of IL-6 were found in LN Class III (545.41 ± 206.32 pg/mL) (*p* = 0.02) and Class IV (429.34 ± 202.43 pg/mL) vs. HC levels (805.45 ± 260.54 pg/mL) (*p* = 0.001).

Basal activity of TAC in LN Class III was found to be significantly decreased (46.30 ± 17.37 µmol) (*p* = 0.03) vs. HC (59.22 ± 7.29 µmol). Basal CAT enzyme activity was found to significantly decrease in LN Class III, 68.13 ± 15.65 KU/mL (*p* = 0.04), and Class IV, 64.36 ± 11.56 KU/mL, in contrast to HC levels, 79.16 ± 6.87 KU/mL (*p* = 0.001). However, SOD enzyme activity was similar between HC, LN Class III, and Class IV Table 3.

The behavior of the oxidants LPO, nitrites/nitrates, and carbonyl groups in basal and six-month follow-up proteins of LN Class III and Class IV were similar.

CRP decreased significantly at six-month follow-up, 4.72 ± 3.23 mg/dL vs. baseline determination in LN Class III (*p* = 0.006). For levels of IL-6 antioxidant enzymes, SOD, CAT, and TAC were similar between baseline and six-month follow-up in NL Class III and LN Class IV patients (Table 4). A graphical summary of the significant results from Table 2 and Table 4 is shown in Figure 2.

## 4. Discussion

SLE is a heterogeneous autoimmune disorder characterized by elevated levels of autoantibodies and multiorgan tissue damage. In SLE, renal involvement develops early in the disease [21]. The inflammatory and oxidative state is a common feature of autoimmune diseases. The action of antioxidants could reduce the severity of the clinical manifestations of LN [22].

In the present study, we aimed to determine the inflammatory and oxidative status of twelve NL Class III and twelve LN Class IV patients recently referred to the Nephrologist. In these patients, serum and plasma were obtained at baseline and six-month follow-up. In addition, the results of twelve subjects who formed the HC group were included, which were used to establish the normal values of the reagents.

Regarding the inflammatory status in this study, IL-6 under-expression was found in LN Class III and Class IV patients at baseline and six-month follow-up compared to the levels found in HC (Table 3). This result was contradictory to some publications that have shown a strong correlation between IL-6 and LN activity [23]. It has also been reported that when cytokine expression decreases, there is a marked reduction of macrophages, CD4+ and CD8+ T lymphocytes infiltrate the kidney, and there is a reduction of IgG, which favors C3 binding in the kidney [24]. This mechanism could explain the decreased expression of IL-6 and CRP in the present study (Table 3). IL-6 is a pleiotropic cytokine that regulates immune and inflammatory responses and affects hematopoiesis, metabolism, and organ development. When IL-6 is deregulated, several autoimmune and inflammatory diseases occur [25]. According to the results obtained in the present study, it is possible to suggest dysregulation of the inflammatory state in patients with LN. Recently, it was published that the state of chronic inflammation contributes significantly to the promotion of OS. OS is associated with increased oxygen burst response in stimulated monocytes/macrophages and neutrophils [26]. However, in a chronic inflammatory state such as LN, there is constant and pathological stimulation of phagocytic cells, resulting in excessive production of ROS. The presence of pro-inflammatory factors in chronic kidney disease (CKD) leads to increased OS Redox imbalance, which favors the alteration of the inflammatory state; therefore, OS and inflammation drive each other. The imbalance in the secretion of pro-inflammatory cytokines increases the expression of Nox4, which is capable of stimulating the synthesis of IL-6 [27]. However, in the present study, the expression of IL-6 was found to be decreased; possibly, the effect of glucocorticoids limits the expression of Nox4.

CRP is widely recognized as a highly conserved acute phase reactant. CRP belongs to the pentraxin family and has five identical non-glycosylated globular subunits. CRP is produced primarily by hepatocytes in response to tissue injury or infection, eliciting an inflammatory response [28]. CRP is expressed in the local inflammatory area where it encounters activated membranes, neutrophil extracellular traps, or acidic pH [29]. In the present study, CRP levels were similar in LN Class III and Class IV patients at baseline determination and HC subjects. CRP levels decreased significantly at six-month follow-up in LN Class III patients (Table 4). This may suggest that the patients were in a controlled inflammatory state or a state of anergy. It is important to consider that CRP levels can increase dramatically after an acute phase stimulus in SLE, although SLE fails to elicit significant CRP production [30], which is consistent with the findings of the present study.

The effect of glucocorticoids on disease management must be considered. Glucocorticoids remain the cornerstone of SLE treatment despite advances in therapeutic protocols and the development of new drugs. The mechanism of action of glucocorticoids is based on reducing the synthesis of pro-inflammatory cytokines, including IL-6, tumor necrosis factor-alpha (TNF-α), and anti-inflammatory cytokines such as IL-37 [31]. The use of glucocorticoids could largely explain the results of IL-6 under-expression found in NL Class III and Class IV patients, or there could be some anergic state in the patients included in the study [32]. In this document, patients were managed mainly with glucocorticoids (methylprednisolone, prednisone), hydrochloroquine, mycophenolic acid, Tacrolimus (one patient (LN Class IV)), and rituximab (1 g per 2 doses and 1 g per 4 doses). Other drugs (antihypertensives, atorvastatin, and calcitriol) were administered as needed.

Relating to OS markers, the significant increase in LPO levels in LN Class III and Class IV patients in contrast to the levels found in HC is striking (Table 3). LPO represents a degradative process as a consequence of the production and propagation of RL reactions involving membrane polyunsaturated fatty acids; LPO is implicated in the pathogenesis of SLE [33]. Increased levels of LPO in the present study could involve key mechanisms for the development of various complications, such as atherosclerosis, Alzheimer’s disease, diabetes, and cancer, and favor end-stage renal disease (ESRD), in addition to increasing SLE activity [34]. MDA is generated in vivo by the peroxidation of polyunsaturated fatty acids and represents a stable end product of lipid peroxidation [35]. OS markers, especially HNE-/MDA-specific immune complexes (ICNs), may be useful in assessing the prognosis of SLE, as well as elucidating the mechanism of disease pathogenesis [36]. An interesting study was recently published in which the authors report that determining free thiol levels could be a better LN biomarker than sRAGE and MDA (main aldehyde of LPO). Furthermore, the authors suggest adding antioxidant agents to current SLE treatment strategies, which could restore redox balance and alleviate some complications induced by SLE [37].

Nitrites/nitrates have been shown to regulate T-cell functions under physiological conditions. Therefore, overproduction of nitrites/nitrates may contribute to T-cell dysfunction [38]. In the present study, basal and six-month follow-up nitrites/nitrates levels did not change in LN Class III, Class IV of NL, and HC.

Carbonyl groups in protein, nitrotyrosine, and oxidized glutathione are stable chemicals used as biomarkers in SLE [39]. The levels of carbonyl groups in proteins from LN Class III and Class IV patients were found to be significantly increased throughout the study, in contrast to the levels obtained in HC. This finding suggests that carbonyl groups in protein actively participate in the oxidative state of the patients throughout the study (Table 3). This is supported by the authors’ publications, where they consider that the increase in carbonyl groups in protein and the induction of serum anti-DNAbc antibodies have an affinity for SLE [40]. The increase in carbonyl groups in protein and LPO could support the important role of OS in LN activity. Carbonyl groups in protein circulate longer in the blood than other oxidized products and are stored for a long time, making them suitable markers for protein oxidation. Several studies have shown elevated levels of carbonyl groups in total protein in patients with SLE [41].

The kidney is one of the vital sources of antioxidant enzymes. When kidney function deteriorates, it is associated with reduced levels of antioxidant enzymes and elevated levels of oxidants. Uremic toxins play a vital role in the occurrence of OS, as in LN. The increase in OS is strictly related to renal and cardiac damage since it aggravates renal dysfunction and induces cardiac hypertrophy. The use of antioxidant therapies may be beneficial by decreasing OS, reducing uremic cardiovascular toxicity, and improving the survival of patients with SLE [42]. In the present study, it is worth noting the significant decrease in the activity of antioxidant enzymes in Class III and Class IV patients included in the study, in contrast to the HC, which could favor the deterioration of cardiac and kidney function.

TAC activity is a non-enzymatic eliminatory system that is an integral part of the defense system responsible for attenuating OS. Reduced TAC has been previously reported in patients with autoimmune diseases such as psoriasis [43]. Accordingly, in the present study, reduced basal levels of TAC were found in LN Class III patients in contrast to HC levels, suggesting an imbalance between antioxidant action and increased oxidants (Table 3). Even chronic systemic inflammation may contribute to the decreased antioxidant activity in SLE patients, predisposing them to develop a higher risk of cardiovascular disease due to the alteration of high-density proteins [44].

CAT is an important endogenous antioxidant enzyme whose main function is characterized by detoxifying hydrogen peroxide (H_2_O_2_) into O_2_ and H_2_O by limiting the deleterious effects of ROS [44]. CAT is considered an essential regulator of OS to chronic exposure to oxidants [45]. Decreased CAT activity affects the oxidant/antioxidant balance, favoring the premature onset of atherogenesis with severe vascular effects in SLE patients by affecting renal blood vessels especially. In the present study, we found a significant decrease in CAT enzyme activity in LN Class III and Class IV patients at baseline, with no change in CAT activity at six-month follow-up, suggesting that patients were exposed to ROS before referral to the Nephrologist and persisted during the six-month follow-up (Table 3).

SOD is an important antioxidant enzyme in the body capable of catalyzing the superoxide anion disproportionation reaction to remove excess ROS from the body [45]. SOD is an active protease containing metallic elements. SOD1, SOD2, and SOD3 have been found in human cells [46]. SOD1 is a metalloenzyme with a superoxide anion detoxifying scavenger effect [47]. SOD exists mainly in the cytoplasmic, peroxidase, and mitochondrial membrane space. SOD1 represents approximately 90% of SOD activity in eukaryotic cells [48]. It was recently reported that serum levels of the antioxidant enzymes Cu/Zn SOD and CAT were significantly reduced in the SLEDAI < to 6 and > to 6 groups of SLE patients compared with controls [14]. However, in our study, SOD activity was similar at baseline and six-month follow-up determination in LN Class III, Class IV, and HC (Table 3 and Table 4). The lack of modification of SOD enzyme activity could be due to the persistence of proteinuria. Proteinuria decreased significantly at the six-month follow-up in LN Class III and Class IV, although SOD activity did not change at the time of the study. On the other hand, elevated levels of circulating anti-SOD2 antibodies were recently detected, which limited the detoxifying activity of SOD2 in LN and membranous nephropathy in association with persistent proteinuria, which could favor worsening renal function [49]. This mechanism could explain why SOD activity in patients with LN was not modified.

Referring to rheumatologic disease outcomes, patients were included according to disease activity reflected by the SLE Disease Activity Index of Disease (SLEDAI) -2K [50]. Striking was the significant decrease in SLEDAI -2K at the six-month follow-up of LN Class III and Class IV patients included in the study, although SLEDAI -2K maintained a high quantification at the end of follow-up (Table 1 and Table 2).

Complement activation is a key event in the pathogenesis of inflammation and tissue injury in SLE [51]. The complement system consists of >30 plasma proteins and cell surface receptors involved in activating and regulating its lytic functions [52]. The involvement of the complement system in SLE has been described over the past 70 years as characterized by low levels of complement proteins C3 and C4. C3 and C4 have potential use for monitoring disease activity and as diagnostic markers for SLE [53]. We analyzed the concentrations of C3 and C4 components, antinuclear antibody (ANA) pattern, and platelet counts in patients with LN (Table 1 and Table 2). In our study, all patients received glucocorticoids with or without immunosuppressive drugs.

The C4 component increased at the six-month follow-up in LN Class III and Class IV patients, although baseline C3 and C4 components were found to be in normal laboratory ranges. However, the classification criteria developed by the American College of Rheumatology (ACR) and the European League Against Rheumatism (EULAR) mention a low plasma complement of (C3, C4, CH50) [54].

In 2019, the EULAR/ACR criteria emphasized the specificity and overall structure of antinuclear antibodies (ANA) as mandatory entry criteria for the diagnosis of SLE and LN [54]. ANAs are a group of autoantibodies that bind to macromolecular components of the cell nucleus. Some ANAs are shown in normal individuals; others are expressed almost exclusively in patients with rheumatic diseases and are useful for diagnosis and prognosis. ANAs are intracellular molecules that are ubiquitously expressed. In SLE, ANAs can promote the development of LN by forming immune complexes that are deposited in the kidney [55]. In the present study, ANAs were found to be significantly increased in LN Class IV patients and decreased non-significantly at six-month follow-up in LN Class III and Class IV patients. The platelet count was not altered throughout the study.

Renal function in LN is mainly characterized by relentless and long-lasting proteinuria. Proteinuria is considered a severe and frequent complication of SLE and a poor prognostic sign [56]. Proteinuria levels are unstable in LN, which is associated with an impaired glomerular filtration membrane self-renewal and repair system [57]. In early LN, it is possible to repair pathological damage to the filtration membrane caused by various complement proteins and cytokines by self-regulation. However, the renal injury cannot be completely repaired. As the disease progresses, severe proteinuria occurs. To prevent proteinuria and repair renal damage in patients with LN, it is urgent to explore the self-repair mechanism of the glomerular filtration membrane in the early stages [58]. In the present study, patients with LN were characterized by proteinuria at Class III and Class IV baseline determination. At the six-month follow-up, proteinuria decreased significantly in both classes without disappearing. Another persistent finding in the present study was hematuria in LN Class III and Class IV patients from baseline, with a significant decrease at six-month follow-up in Class III (Table 2). Persistent hematuria could be due to a common vascular affectation occurring in LN. The mechanism of persistent hematuria could be explained by uncomplicated vascular immune deposits that form common vasculopathy. Another less common mechanism is real vasculitis, which requires determination of morphology by immunofluorescence [59].

The glomerular filtration rate (GFR) during the whole study was found to be in normal values at baseline and six-month follow-up, as previously reported [60] (Table 1 and Table 2). Limited data on LN outcomes with moderate to severe renal impairment at presentation (defined by a low GFR <30 mL/min) exist. There is evidence that proteinuria and hematuria do not accurately measure the direction or magnitude of change in GFR in LN. [61]. In SLE disease, there are multiple alterations in addition to the markers that were evaluated in the present study. Increased oxidative stress can contribute to the pathogenesis of SLE by generating oxidative modifications in proteins, lipids, and DNA, which lead to the deregulation of the immune system and trigger autoimmune attacks. This occurs through mechanisms such as enhanced NETosis, activation of the mTOR pathway, and unbalanced T-cell differentiation. Close monitoring of patients favors the timely detection and management of inflammatory and oxidative status with the purpose of determining an accurate assessment of the degree of OS [62]. Therefore, pharmacological approaches targeting OS offer promising results in the treatment of patients with SLE.

## 5. Conclusions

In the present study, patients showed an imbalance in oxidative status from the beginning of the diagnosis of LN. The imbalance of the oxidative state was characterized by the increase in the oxidants LPO and protein carbonyl groups and the decrease in the activity of the antioxidant enzymes TAC and CAT compared to HC. However, the patients did not present an increase in disease activity and renal function improvement. Renal function of LN Class III and Class IV patients showed a significant decrease in proteinuria and hematuria without achieving negativization at six-month follow-up. The glomerular filtration rate did not change during the length of the study. SLEDAI -2K decreased in both classes of patients at six-month follow-up, and C3 and C4 increased at the end of the study. The early co-management between Rheumatologists and Nephrologists is essential to prevent the rapid progression of LN. It would be interesting to administer antioxidant supplements to patients with a recent diagnosis of LN and evaluate its effect in a follow-up study.

Study strengths. The originality of the study is based on the inclusion of patients recently diagnosed with Class III and Class IV LN with a six-month follow-up where inflammatory markers (IL-6 and CRP), three oxidative stress markers (LPO, carbonyl groups in proteins and nitrites/nitrates), and the antioxidant activity of enzymes (SOD, CAT, TAC) contrasted to the found in HC were determined.

Limitations of the study. A limited number of patients with systemic lupus erythematosus with recent detection of renal alterations were included. Patient follow-up time was short. All patients continued to receive their SLE treatment, which consisted primarily of glucocorticoids. However, the type of glucocorticoid and doses prescribed may vary according to the response characteristics of each patient.

## Figures and Tables

**Figure 1 antioxidants-12-02065-f001:**
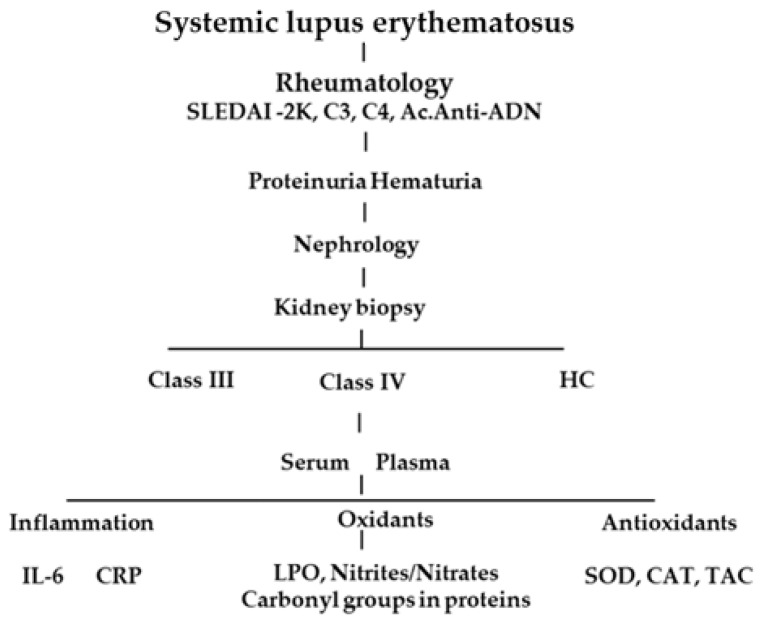
Project flowchart.

**Figure 2 antioxidants-12-02065-f002:**
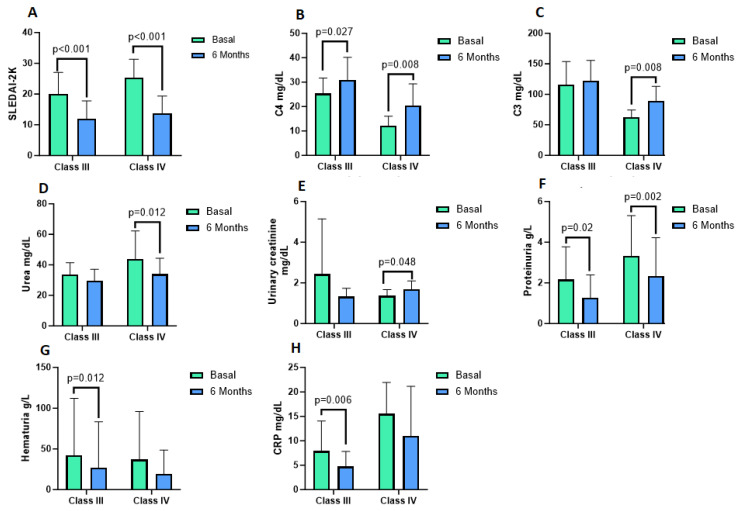
Significant results for markers of LN disease and inflammation after the 6-month follow-up. (**A**–**C**) show systemic lupus erythematosus data. (**D**–**G**) are markers of renal function. (**H**) shows a marker for inflammation. CRP = C-reactive protein, C3 = complement protein C3, C4 = complement protein C4.

**Table 1 antioxidants-12-02065-t001:** Basal anthropometric and biochemical data in Class III and Class IV of lupus nephritis.

	Class III N-12	Class IV N-12	*p*
Gender			
Female n (%)	10 (83.3)	11 (91.7)	0.55
Male n (%)	2 (16.7)	1 (8.3)	
Arterial Hypertension n%	3 (25)	3 (25)	1
Age years	27.83 ± 7.06	33.00 ± 7.87	0.11
Weight Kg	68.48 ± 23.26	63.21 ± 15.64	0.52
Height cm	163.42 ± 10.80	155.92 ± 4.46	0.037
BMI Kg/m^2^	24.95 ± 1.26	25.66 ± 1.47	0.72
Systemic lupus erythematosus data			
SLEDAI -2K	20.08 ± 6.92	25.33 ± 6.01	0.060
C3 mg/dL	115.97 ± 37.80	62.82 ± 11.40	<0.001
C4 mg/dL	25.36 ± 6.34	12.18 ± 3.90	<0.001
Anti-DNA IU/mL	25.12 ± 32.75	73.69 ± 74.69	0.05
Biochemical data			
Hemoglobin g/dL	12.17 ± 2.40	11.23 ± 1.98	0.31
Hematocrit g/dL	35.04 ± 6.50	35.76 ± 5.41	0.77
Platelets thousands/uL	260.92 ± 64.19	239.92 ± 98.87	0.54
Leukocytes thousands/uL	7.67 ± 3.31	7.46 ± 4.24	0.89
Glucose mg/dL	90.76 ± 10.26	98.77 ± 12.34	0.09
Albumin mg/dL	3.54 ± 1.00	3.40 ± 0.69	0.69
Chlorine mm/L	110.17 ± 6.82	107.58 ± 2.87	0.24
Potassium mm/L	4.33 ± 1.03	4.58 ± 0.97	0.56
Phosphorus mm/L	4.08 ± 0.80	3.87 ± 0.82	0.54
Calcium mm/L	2.24 ± 0.26	2.32 ± 0.17	0.42
Sodium mm/L	137.75 ± 3.05	138.50 ± 3.15	0.56
Magnesium mm/L	0.78 ± 0.13	0.83 ± 0.09	0.28
Renal function			
Glomerular filtration rate mL/min/1.73 m^2^	79.23 ± 30.97	104.48 ± 40.85	0.10
Urea mg/dL	33.60 ± 7.82	43.90 ± 18.35	0.09
Creatinine mg/dL	1.16 ± 0.44	0.87 ± 0.51	0.15
Proteinuria g/L	2.16 ± 1.62	3.33 ± 1.98	0.13
Albuminuria mg/24 h	64.88 ± 88.61	126.60 ± 120.85	0.17
Urinary creatinine mg/dL	2.43 ± 2.71	1.36 ± 0.30	0.19
Hematuria g/L	42.25 ± 69.89	37.08 ± 58.81	0.85

Values are expressed with frequency (percentage) or mean ± standard deviation (SD). Chi^2^ was used for dichotomous variables, and independent sample *t*-test was used for quantitative variables. BMI = body mass index, C3 = complement protein C3, C4 = complement protein C4.

**Table 2 antioxidants-12-02065-t002:** Biochemical and renal function basal and six-month follow-up data in Class III and Class IV of lupus nephritis.

	Class III N-12			Class IV N-12		
	Baseline	Six-Month Follow-Up	*p*	Baseline	Six-Month Follow-Up	*p*
Systemic lupus erythematosus data						
SLEDAI -2K	20.08 ± 6.92	11.92 ± 5.87	<0.001	25.33 ± 6.01	13.83 ± 5.52	<0.001
C3 mg/dL	115.97 ± 37.80	122.43 ± 33.26	0.54	62.82 ± 11.40	89.64 ± 23.22	0.008
C4 mg/dL	25.36 ± 6.34	30.91 ± 9.22	0.027	12.18 ± 3.90	20.33 ± 8.95	0.008
Anti DNA IU/mL	25.12 ± 32.75	18.33 ± 17.18	0.23	73.69 ± 74.69	41.72 ± 27.42	0.06
Biochemical data						
Hemoglobin g/dL	12.17 ± 2.40	13.00 ± 1.63	0.12	11.23 ± 1.98	12.13 ± 1.34	0.08
Hematocrit %	35.04 ± 6.50	38.57 ± 7.94	0.28	35.76 ± 5.41	39.29 ± 6.75	0.009
Platelets thousands/uL	260.92 ± 64.19	266.67 ± 68.12	0.73	239.92 ± 98.87	269.33 ± 72.79	0.22
Leukocytes	7.67 ± 3.31	7.79 ± 2.28	0.89	7.46 ± 4.24	7.99 ± 2.18	0.74
Glucose mg/dL	90.76 ± 10.26	89.06 ± 8.75	0.56	98.77 ± 12.34	82.45 ± 9.82	0.005
Albumin mg/dL	3.54 ± 1.00	3.67 ± 0.72	0.66	3.40 ± 0.69	3.52 ± 1.00	0.61
Chlorine mmol/L	110.17 ± 6.82	103.92 ± 4.78	0.007	107.58 ± 2.87	105.00 ± 3.33	0.05
Potassium mm/L	4.33 ± 1.03	4.41 ± 0.51	0.76	4.58 ± 0.97	4.40 ± 0.42	0.42
Phosphorus mm/L	4.08 ± 0.80	3.41 ± 1.36	0.19	3.87 ± 0.82	3.27 ± 1.45	0.35
Calcium mmol/L	2.24 ± 0.26	2.35 ± 0.15	0.14	2.32 ± 0.17	2.23 ± 0.16	0.15
Sodium mmol/L	137.75 ± 3.05	138.33 ± 1.23	0.56	138.50 ± 3.15	140.25 ± 3.25	0.09
Magnesium mmol/L	0.78 ± 0.13	0.81 ± 0.12	0.27	0.83 ± 0.09	0.83 ± 0.11	0.85
Renal function						
Glomerular filtration rate mL/min/1.73 m^2^	79.23 ± 30.97	88.55 ± 17.95	0.26	104.48 ± 40.85	104.44 ± 26.47	0.99
Urea mg/dL	33.60 ± 7.82	29.65 ± 7.46	0.23	43.90 ± 18.35	33.93 ± 10.46	0.012
Creatinine mg/dL	1.16 ± 0.44	0.97 ± 0.25	0.15	0.87 ± 0.51	0.79 ± 0.27	0.47
Proteinuria g/L	2.16 ± 1.62	1.26 ± 1.13	0.02	3.33 ± 1.98	2.33 ± 1.89	0.002
Albuminuria mg/24 g	64.88 ± 88.61	20.68 ± 20.71	0.08	126.60 ± 120.85	138.92 ± 382.82	0.90
Urinary creatinine mg/dL	2.43 ± 2.71	1.16 ± 1.49	0.68	1.36 ± 0.30	1.69 ± 0.41	0.048
Hematuria g/L	42.25 ± 69.89	27.00 ± 56.41	0.012	37.08 ± 58.81	19.42 ± 29.11	0.11

Values are expressed as mean ± standard deviation (SD); a paired-sampled *t*-test was used. C3 = complement protein C3, C4 = complement protein C4.

**Table 3 antioxidants-12-02065-t003:** Inflammation and oxidative stress markers in NL Class III and Class IV vs. HC.

	HC N-12	Class III N-12	HC vs. Class III *p*	Class IV N-12	HC vs. Class IV *p*	Class III vs. Class IV *p*
Inflammation markers						
CRP mg/dL	7.93 ± 6.14	4.72 ± 3.08	0.09	15.53 ± 6.45	10.99 ± 10.17	0.08
IL-6 pg/mL	805.45 ± 260.54	545.41 ± 206.32	0.02	429.34 ± 202.43	0.001	0.19
Oxidants						
Lipoperoxides mM	0.06 ± 0.01	0.54 ± 0.11	0.0001	0.51 ± 0.12	0.0001	0.64
Nitrites /nitrates µg/mL	0.45 ± 0.25	0.44 ± 0.19	0.89	0.38 ± 0.09	0.39	0.39
Carbonyl groups in proteins µmol	0.45 ± 0.25	4.08 ± 0.58	0.0001	4.29 ± 0.47	0.0001	0.35
Antioxidants						
Total antioxidant capacity micromol trolox equivalents	59.22 ± 7.29	46.30 ± 17.37	0.03	53.99 ± 9.97	0.16	0.19
Catalase KU/mL	79.16 ± 6.87	68.13 ± 15.65	0.04	64.36 ± 11.56	0.001	0.51
Superoxide dismutase U/mL	0.01 ± 0.01	0.02 ± 0.01	0.29	0.019 ± 0.01	0.10	0.42

Values are expressed as mean ± standard deviation (SD). Independent sample *t*-test was used. CRP = C-reactive protein, IL-6 = interleukin 6.

**Table 4 antioxidants-12-02065-t004:** Oxidative stress markers in SLE, Class III, and Class IV with lupus nephritis vs. baseline and six-month follow-up.

	Class III N-12			Class IV N-12		
	Baseline	Six-Month	*p*	Baseline	Six-Month	*p*
Inflammation markers						
CRP mg/dL	7.93 ± 1.77	4.72 ± 3.23	0.006	15.53 ± 6.45	10.99 ± 10.17	0.21
IL-6 pg/mL	545.41 ± 206.32	462.84 ± 228.87	0.56	429.34 ± 202.42	471.21 ± 242.43	0.79
Oxidants						
Lipoperoxides	0.54 ± 0.11	0.49 ± 0.16	0.20	0.51 ± 0.12	0.52 ± 0.09	0.61
Nitrites/nitrates	0.44 ± 0.19	0.42 ± 0.18	0.65	0.38 ± 0.09	0.43 ± 0.08	0.29
Carbonyl groups in proteins µmol	4.08 ± 0.58	5.16 ± 4.5	0.38	4.29 ± 0.47	4.42 ± 1.59	0.78
Antioxidants						
Total antioxidant capacidad micromol trolox equivalents	46.30 ± 17.37	52.73 ± 8.95	0.26	53.99 ± 9.97	54.57 ± 16.17	0.93
Catalase KU/mL	68.13 ± 15.65	66.72 ± 12.72	0.76	64.36 ± 11.56	65.72 ± 12.13	0.71
Superoxide dismutase U/mL	0.02 ± 0.01	0.02 ± 0.01	0.18	0.02 ± 0.01	0.02 ± 0.01	1.00

Values are expressed as mean ± standard deviation (SD). A paired-sampled *t*-test was used. CRP = C-reactive protein, IL-6 = interleukin 6.

## Data Availability

The database that supports the conclusions of this research work will be made available by the authors upon express request and with the authorization of the Ethics and Research Committee.

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
