# Peer review of "The Inflammatory and Oxidative Status of Newly Diagnosed Class III and Class IV Lupus Nephritis, with Six-Month Follow-Up"

_antioxidants, 2023, doi:10.3390/antiox12122065_

Round 1

Reviewer 1 Report

Comments and Suggestions for Authors

This is a prospective study in 24 patients with active lupus nephritis and 12 healthy individuals, followed for 6 months, to determine changes in CRP, IL-6 and oxidative stress/anti-oxidants in the serum. There are several points to be considered by the Authors before this work is acceptable for publication.

1. I found the abstract very confusing. I would suggest a clearer way of data presentation. Also, the final conclusive statement makes no sense in view of the results/scope of the work.

2. Details on patient recruitment, inclusion and exclusion criteria should be added.

3. Was treatment allowed at the time of enrollment/sampling? Judging from the Discussion, I guess yes. But, this might have introduced bias in the recordings, especially inflammatory indices which - as discussed by the authors - may be affected by administered therapies such as glucocorticoids. To this end, the finding of "reduced" IL-6 as compared to HC, is unexpected.

4. Determination of antioxidants. I am wondering whether the measured levels of anti-oxidants should be adjusted to the eGFR level of the patients. What is known about the renal clearance of SOD, catalase etc.? Could this have influenced the results at baseline vs. 6 months post-therapy?

5. Statistical methods. I am not sure the appropriate tests have been performed. Paired-samples analysis is not done with Chi-squared test, rather with McNemar's test. Same for t-test; it is not clear whether paired-sampled t-test was used.

6. Tables. Adjusted for multiple comparisons should be applied. Otherwise, some of the "significant" associations may be false-positives.

7. I found the Discussion very extensive and thus, should be shortened. Especially some parts are not relevant to the scope of the work (eg reference to recommnedations etc).

Comments on the Quality of English Language

Minor edits should be done. 

Author Response

Reviewer 1

This is a prospective study in 24 patients with active lupus nephritis and 12 healthy individuals, followed for 6 months, to determine changes in CRP, IL-6 and oxidative stress/anti-oxidants in the serum. There are several points to be considered by the Authors before this work is acceptable for publication.

Comment. 1. I found the abstract very confusing. I would suggest a clearer way of data presentation. Also, the final conclusive statement makes no sense in view of the results/scope of the work.

Answer. We agree with the comment. The summary was improved to read as follows:

Abstract: Lupus nephritis (LN) is the most frequent and severe complication of systemic lupus erythematosus (SLE). A prospective cohort with a six-month follow-up was performed. Twelve SLE patients diagnosed with LN Class III, twelve NL Class IV patients, and twelve healthy control subjects (HC) were included. SLE data, renal function, oxidants, antioxidants, and inflammation were determined at baseline and six-month follow-up. During the six-month follow-up, the SLE Disease Activity Index (SLEDAI 2K) decreased in both LN Class III (20.08±6.92 vs. 11.92±5.87, p<0.001) and LN Class IV (25.33±6.01 vs. 13.83±5.52, p<0.001) patients. Furthermore, the values of the C4 component increased during follow-up also for LN Class III (25.36±6.34 vs. 30.91±9.22, p=0.027) and LN Class IV (12.18±3.90 vs. 20.33±8.95, p=0.008) groups. Regarding inflammation markers, both groups presented decreased C-reactive protein (CRP) but were significant only for patients with LN Class III (7.93±1.77 vs. 4.72±3.23, p=0.006). Renal function remained stable in both groups with no changes in eGFR. Patients with LN Class III and Class IV showed higher baseline levels for lipoperoxides (Class III p<0.01, Class IV, p<0.1) and carbonyl groups in proteins (Class III p<0.01, Class IV p<0.1) compared to HC. Besides, both groups presented lower baseline values of Total antioxidant capacity (Class III p<0.01, Class IV p<0.1) and catalase (Class III p<0.01, Class IV p<0.1) compared to HCs. However, antioxidant and oxidant markers did not show significant differences between baseline values and at six months for either of the two study groups.

In conclusion, during the first six months of LN diagnosis, Class III and Class IV patients treated for SLE do not present an increase in disease activity or renal function. There was no increase in oxidative stress in the early months after the diagnosis of LN. Nevertheless, the decrease in inflammation markers may represent a warning signal to continue monitoring to prevent oxidative stress from contributing to the development of the disease in the future.

Conclusions

The study shows that during the first six months of LN diagnosis, Class III and Class IV patients treated for SLE do not present an increase in disease activity. This is observed with the increase in complement C4 values at six months of follow-up. Furthermore, the increase in C4 was also accompanied by a decrease in the SLEDAI 2K score. Similarly, kidney function appeared to be stable during the study. The eGFR remained unchanged, with a decrease in proteinuria and hematuria. On the other hand, patients with LN showed a baseline decrease in antioxidants and an increase in oxidants compared to HC. However, the six months of follow-up were insufficient to observe significant changes in OS markers. Even so, during this time, an increase in inflammation (CRP) was found in Class III patients. The early co-management between Rheumatologists and Nephrologists is essential to prevent the rapid progression of LN. Chronic diseases such as LN should be carefully monitored to prevent factors such as OS or inflammation from aggravating the disease.

Comment. 2. Details on patient recruitment, inclusion, and exclusion criteria should be added.

Answer: We considered the comment and modified the Material and Methods section as shown below:

Women and men over 18 years old with SLE and a recent diagnosis of focal lupus nephritis (LN Class III) or diffuse lupus nephritis (LN Class IV) were included. The patients were selected from the Department of Nephrology of the Specialty Hospital of the National Medical Center of the West of the Mexican Institute of Social Security (IMSS) in Guadalajara, Jalisco, Mexico. The included patients agreed to participate in the study. All patients maintained their SLE treatment which consisted primarily of glucocorticoids. Patients who ingested antioxidants (vitamin E, vitamin C, etc.) three months before the study or patients with clinical or biochemical data of an infectious process were not included. Patients with any type of diabetes, with a history of thrombotic events or polycystic diseases were also excluded. In addition, patients who withdrew Informed Consent were excluded from the study.

3. Comment. Was treatment allowed at the time of enrollment/sampling? Judging from the Discussion, I guess yes. However, this might have introduced bias in the recordings, especially inflammatory indices which - as discussed by the authors - may be affected by administered therapies such as glucocorticoids. To this end, the finding of "reduced" IL-6 as compared to HC, is unexpected.

Answer. All patients maintained their SLE treatment which consisted primarily of glucocorticoids. Treatment for SLE was a homogeneous factor in both study groups. However, we are aware that the type of glucocorticoid and dosage may vary depending on the needs of each patient. For these reasons, we modified the limitations of the study as follows:

Limitations of the study. A limited number of patients with systemic lupus erythematosus with recent detection of renal alterations were included. Patient follow-up time was short. All patients continued to receive their SLE treatment which consisted primarily of glucocorticoids. However, the type of glucocorticoid and doses prescribed may vary according to the response characteristics of each patient.

4. Comment. Determination of antioxidants. I am wondering whether the measured levels of anti-oxidants should be adjusted to the eGFR level of the patients. What is known about the renal clearance of SOD, catalase, etc.? Could this have influenced the results at baseline vs. 6 months post-therapy?

Answer. In the present study, eGFR was maintained without difference between the two groups at baseline and six-month follow-up, so it was not considered an intervening factor. However, we performed a correlation analysis to determine the correspondence between oxidative stress markers and eGFR, finding no significant correlation for these patients. It should be noted that during the time of the study, patients maintained healthy eGFR values

Results of the Pearson correlation analysis between eGFR and oxidative stress markers. The data is not displayed. If you think these results should be included, we will include them without a problem.

Correlations

eGFR basal

eGFR 6M

r

p

r

p

Lipoperoxides

-0.290

0.169

0.050

0.822

/nitrates

0.157

0.475

0.139

0.517

IL-6

-0.164

0.454

-0.194

0.374

Total antioxidant capacity

0.384

0.064

-0.111

0.604

Catalase

0.280

0.184

0.157

0.465

Superoxide dismutase

-0.176

0.410

-0.094

0.662

Carbonyl groups in proteins

0.086

0.704

-0.066

0.775

5. Comment. Statistical methods. I am not sure the appropriate tests have been performed. Paired-sample analysis is not done with the Chi-squared test, rather with McNemar's test. Same for t-test; it is not clear whether paired-sampled t-test was used.

Answer. We appreciate the comment. Certainly, McNemar's test is suitable for Paired-samples analysis. For this study, McNemar's test could be used to compare dichotomous variables between the baseline measurement and at six months. However, in the present study, only the baseline dichotomous variables between the two study groups (Class III and Class IV) were analyzed. As these study groups are considered independent samples, the Chi-squared test was considered a better analysis option.

For the use of the t-test, we clarify that the independent sample t-test was used to compare the data between the two study groups (Tables 1 and 3). The paired-sampled t-test was used for comparison between baseline values and at six months for each study group (Tables 2 and 4). The table footers were modified to accommodate the previous clarifications.

6. Comment. Tables. Adjusted for multiple comparisons should be applied. Otherwise, some of the "significant" associations may be false positives.

Answer. We take into consideration the comment. We also mention that Tables 1 and 2 show a simple comparison between two independent and dependent samples respectively, so we do not consider adjustment for multiple comparisons necessary. Same case for table 4. On the other hand, Table 3 shows values from three independent samples. However, significant results of Table 3 were also obtained using simple comparison analysis. We modified the table footers to clarify the analyses used.

7. Comment. I found the Discussion very extensive and thus, should be shortened. Especially some parts are not relevant to the scope of the work (eg reference to recommendations etc).

Answer. I apologize for the excess text. Appropriate changes to improve the writing in the discussion section were made.

Comment. Comments on the Quality of English Language Minor edits should be done. 

Answer. We review the document to correct language errors. We expect the manuscript to meet the requested improvements

Reviewer 2 Report

Comments and Suggestions for Authors

The article is interesting and raises important aspects.
It is known that Systemic lupus erythematosus (SLE) is a heterogeneous autoimmune disease associated with severe organ damage. It is known about its inflammatory and oxidative stress-related relationships. There have been publications on this subject recently, like  Liu, L.; de Leeuw, K.; Arends, S.; Doornbos-van der Meer, B.; Bulthuis, M.L.C.; van Goor, H.; Westra, J., on behalf of the Dutch LN Studies. Biomarkers of Oxidative Stress in Systemic Lupus Erythematosus Patients with Active Nephritis. Antioxidants 2023, 12, 1627. https://doi.org/10.3390/antiox12081627  - on a larger, if compared to this study group of patients with LN, have done their study and concluded that their results indicated that oxidative stress levels in LN patients are increased compared to HC and associated with SLE disease activity. They also found that the interventional therapy to restore redox homeostasis may be useful as an adjunctive therapy in treating oxidative damage in SLE.
The authors did not mention this publication and many similar ones.

I have also a few comments:

The authors wrote high specificity CRP on line 116 - this should be corrected to high sensitivity CRP.
In turn, in section 2.3.1 they wrote C-reaction protein - and I wonder whether they meant CRP or hsCRP?

I recommend reading the work to correct many errors - type CuCl2 instead of CuCl_2; or NH4Ac instead of NH_4Ac.

I have a question about height and weight. Maybe authors should calculate your BMI? I am not convinced by the information that some patients were significantly shorter than patients from the second group; after all, so what?

In general, the article is very difficult to read; there are no figures and too much text. This should be corrected.

I have also a question concerning the study group. Was someone excluded? What about comorbid conditions?
Conclusion should be reformulated, they are rather weak in the current version.
Who is HC? And how they were recruited? Are there healthy volunteers?
A flow diagram should be here useful.
What is the novelty of this article?

Comments on the Quality of English Language

Minor editing of English language is required. Many small errors.

Author Response

Reviewer 2

Comment. The article is interesting and raises important aspects.

It is known that Systemic lupus erythematosus (SLE) is a heterogeneous autoimmune disease associated with severe organ damage. It is known about its inflammatory and oxidative stress-related relationships. There have been publications on this subject recently, like Liu, L.; de Leeuw, K.; Arends, S.; Doornbos-van der Meer, B.; Bulthuis, M.L.C.; van Goor, H.; Westra, J., on behalf of the Dutch LN Studies. Biomarkers of Oxidative Stress in Systemic Lupus Erythematosus Patients with Active Nephritis. Antioxidants 2023, 12, 1627. https://doi.org/10.3390/antiox12081627 - on a larger, if compared to this study group of patients with LN, have done their study and concluded that their results indicated that oxidative stress levels in LN patients are increased compared to HC and associated with SLE disease activity. They also found that the interventional therapy to restore redox homeostasis may be useful as an adjunctive therapy in treating oxidative damage in SLE.

The authors did not mention this publication and many similar ones.

Answer. The recent and interesting information provided by the article published by Liu L, et al., was included in the discussion section. The importance of determining the behavior of antioxidants and oxidants in SLE and LN is clear. Our document provides more information on this matter. Thank you for the recommendation of such a demonstrative article.

Discussion section: An interesting study was recently published in which the authors report that determining free thiol levels could be a better biomarker in LN compared to sRAGE and MDA (main aldehyde of LPO). Furthermore, the authors suggest adding antioxidant agents to current SLE treatment strategies, which could restore redox balance and alleviate some complications induced by SLE. Reference 44

I have also a few comments:

Comment. The authors wrote high specificity CRP on line 116 - this should be corrected to high sensitivity CRP.

Answer. High-sensitivity CRP is correct

Comment. In turn, in section 2.3.1 they wrote C-reaction protein - and I wonder whether they meant CRP or hsCRP?

Answer. CRP is correct

Comment. I recommend reading the work to correct many errors - type CuCl2 instead of CuCl_2; or NH4Ac instead of NH_4Ac.

Answer. CuCl2 and NH4Ac is correct

Comment. I have a question about height and weight. Maybe authors should calculate your BMI? I am not convinced by the information that some patients were significantly shorter than patients from the second group; after all, so what?

Answer. Patients of NL Class IV are shorter of NL Class III

BMI in Class III are 24.95 ± 1.26 kg/m2 and Class IV are 25.66 ± 1.47 kg/m2  p=0.72 (Table1)

Comment. In general, the article is very difficult to read; there are no figures and too much text. This should be corrected.

Answer. I apologize for the excess text. Many results were modified from the initial determination to the final result. We hope that the modifications in the text are sufficient

We separated the database into anthropometric results, data referring to SLE, biochemical data, renal function data, and data referring to inflammatory and oxidative status.

Therefore, we consider the modifications that were observed appropriate to discuss. We can split the tables, report the results, and discuss according to each table

Comment. I have also a question concerning the study group.  Was someone excluded?

Answer. No patient was excluded from baseline determination until six months of follow-up. One Class IV patient died after one year. The cause of death was hypovolemic shock caused by gastrointestinal bleeding

Comment. What about comorbid conditions?

Answer. Three (25%) patients from Class III and three (25%) patients from Class IV of LN are hypertensive (p=1). The data is included in Table 1.

Comment. Conclusion should be reformulated, they are rather weak in the current version.

Answer.

Conclusion

The study shows that during the first six months of LN diagnosis, Class III and Class IV patients treated for SLE do not present an increase in disease activity. This is observed with the increase in complement C4 values at six months of follow-up. Furthermore, the increase in C4 was also accompanied by a decrease in the SLEDAI 2K score. Similarly, kidney function appeared to be stable during the study. The eGFR remained unchanged, with a decrease in proteinuria and hematuria. On the other hand, patients with LN showed a baseline decrease in antioxidants and an increase in oxidants compared to HC. However, the six months of follow-up were insufficient to observe significant changes in OS markers. Even so, during this time, an increase in inflammation (CRP) was found in Class III patients. The early co-management between Rheumatologists and Nephrologists is essential to prevent the rapid progression of LN. Chronic diseases such as LN should be carefully monitored to prevent factors such as OS or inflammation from aggravating the disease.

Comment. Who is HC? And how they were recruited? Are there healthy volunteers?

Answer. HC is a healthy control subject.

Ten mL were obtained from 12 extra blood donors who came to the blood bank and who agreed to donate an extra 10 mL of blood. These subjects formed the healthy control (HC) group. The HC samples were used to establish the normal value of the reagents to determine the inflammatory and oxidative status in NL.

Comment. A flow diagram should be here useful.

Answer. The flow chart of the research work is shown. If you consider it should be included in the main document, we will gladly add it

 Comment. What is the novelty of this article?

Answer. The originality of the study is based on the inclusion of patients recently diagnosed with Class III and Class IV LN with a six-month follow-up where inflammatory markers (IL-6 and CRP), three oxidative stress markers (LPO, carbonyl groups in proteins and nitrites/nitrates) and the antioxidant activity of enzymes (SOD, CAT, TAC) contrasted front the found in HC were determined

Comment. Minor editing of the English language is required. Many small errors.

Answer. We review the document to correct language errors. We hope there are many fewer

Round 2

Reviewer 1 Report

Comments and Suggestions for Authors

I have no further suggestions or comments. 

Author Response

Dear reviewer

On behalf of the authors of the research, I thank you for your comments. I know that the comments helped to improve the initial document

Alejandra Guillermina Miranda-Díaz, MD, PhD

Reviewer 2 Report

Comments and Suggestions for Authors The article sent after the corrections is difficult to evaluate because the authors did not indicate (highlighted) changes they made in the text.

Therefore, it is challenging to follow the changes introduced.
I read the article carefully and admit that I do not understand some of the changes. In my opinion, the conclusions still have shortcomings. Authors should avoid phrases like "an increase in disease activity or renal function." I believe such a statement is incorrect; it is better to write an "increase in disease activity and renal function improvement." The sentence in the abstract - I don't understand it at all: "Nevertheless, the decrease in inflammation markers may represent a warning signal to continue monitoring to prevent OS from contributing to the development of the disease in the future." What do the authors mean - a decrease in inflammatory biomarkers for further monitoring is a warning signal????

The article is generally fascinating, and a lot of important research has been done, but in my opinion, it is still not suitable for publication in its current version. The discussion needs to change. It can break down inflammation and oxidative stress into small subsections. What the authors write in the discussion is often quite simple and obvious.

I would also like to refer to the literature; many of the publications the authors refer to are over 15 years old, which is not the best, especially in bio-med research. For example, today, we know much more about CRP and oxidative stress. I propose to remove the obvious and deepen the discussion about mechanisms.

The conclusions in the abstract are different from the conclusions in the article. But they are still not good; I would even say they are, in some parts, even wrong.

The authors aspire to publish in a perfect journal focused on oxidative stress. In my opinion, in this article, oxidative stress is only an assessment of biomarkers - but without any more profound reflection on what results from it.

I suggest the authors consider this. They should focus on the articles on oxidative stress in kidney diseases; there is no shortage of them
They should write in the article about the novelty of their results, and consider what their article brings:

Let's look:
Antioxidants 2020, 9(8), 752; https://doi.org/10.3390/antiox9080752
Antioxidants 2020, 9(11), 1079; https://doi.org/10.3390/antiox9111079
J Inflamm Res. 2023 Feb 4;16:453-465. doi: 10.2147/JIR.S399284. PMID: 36761905; PMCID: PMC9907008.
Antioxidants 2023, 12(8), 1627; https://doi.org/10.3390/antiox12081627
Lupus. 2020;29(3):311-323. doi:10.1177/0961203320904784

As before I have suggested, I also recommend inserting a flow diagram. The authors wrote: "The flow chart of the research work is shown. If you consider it should be included in the main document, we will gladly add it." I don't know where it is visible, but please include it in the publication.

Finally, I think it would be helpful to have a chart/figure  showing changes in parameters over time in different groups, which would show the course of changes and make it easier to follow them.

Author Response

Reviewer 2

Comment. The article sent after the corrections is difficult to evaluate because the authors did not indicate (highlighted) changes they made in the text.

Answer. The document is attached with the changes highlighted

Comment. Therefore, it is challenging to follow the changes introduced.
I read the article carefully and admit that I do not understand some of the changes. In my opinion, the conclusions still have shortcomings.

Answer. We made improvements to the conclusion, showing as follows:

Conclusion section:

In the present study, patients show an imbalance in oxidative status from the beginning of the diagnosis of LN. The imbalance of the oxidative state was characterized by the increase in the oxidants LPO and protein carbonyl groups and the decrease in the activity of the antioxidant enzymes TAC and CAT compared to HC. However, the patients did not present an increase in disease activity or renal function during follow-up. Renal function of LN Class III and Class IV patients showed a significant decrease in proteinuria and hematuria without achieving negativization at six-month follow-up. The glomerular filtration rate did not change during the length of the study. SLEDAI -2K decreased in both classes of patients at six-month follow-up, and C3 and C4 increased at the end of the study. The early co-management between Rheumatologists and Nephrologists is essential to prevent the rapid progression of LN. It would be interesting to administer antioxidant supplements to patients with a recent diagnosis of LN and evaluate its effect in a follow-up study. (lines 481-491)

Comment. Authors should avoid phrases like "an increase in disease activity or renal function." I believe such a statement is incorrect; it is better to write an "increase in disease activity and renal function improvement."

Answer. I appreciate the comment. The phrase in the abstract was corrected. “an increase in disease activity and renal function improvement” (Lines 45, 484 and 485)

Comment. The sentence in the abstract - I don't understand it at all:"Nevertheless, the decrease in inflammation markers may represent a warning signal to continue monitoring to prevent OS from contributing to the development of the disease in the future." What do the authors mean - a decrease in inflammatory biomarkers for further monitoring is a warning signal????

Answer. We made improvements to the abstract conclusion, showing as follows:

In conclusion, patients show an imbalance in the oxidative state characterized by the increase in the oxidants LPO and protein carbonyl groups and the decrease in the activity of the antioxidant enzymes TAC and CAT compared to HC. However, the patients did not present an increase in disease activity and renal function improvement. The glomerular filtration rate did not change during the length of the study, and SLEDAI -2K, C3, and C4 improved. The early co-management between Rheumatologists and Nephrologists is essential to prevent the rapid progression of LN. It would be interesting to administer antioxidant supplements to patients with a recent diagnosis of LN and evaluate its effect in a follow-up study. (lines 42-49)

Comment. The article is generally fascinating, and a lot of important research has been done, but in my opinion, it is still not suitable for publication in its current version. The discussion needs to change. It can break down inflammation and oxidative stress into small subsections. What the authors write in the discussion is often quite simple and obvious. I would also like to refer to the literature; many of the publications the authors refer to are over 15 years old, which is not the best, especially in bio-med research. For example, today, we know much more about CRP and oxidative stress. I propose to remove the obvious and deepen the discussion about mechanisms.

Answer. The Suggested references are greatly appreciated. The discussion of some of the findings was improved with information on mechanisms related to oxidative stress, SLE, and LN. The changes are highlighted in blue on lines 327-336, 366-370, 386-393, 467-478.

Comment. The conclusions in the abstract are different from the conclusions in the article. But they are still not good; I would even say they are, in some parts, even wrong.

Answer. We made improvements to the abstract conclusion, showing as follows:

In conclusion, patients show an imbalance in the oxidative state characterized by the increase in the oxidants LPO and protein carbonyl groups and the decrease in the activity of the antioxidant enzymes TAC and CAT compared to HC. However, the patients did not present an increase in disease activity and renal function improvement. The glomerular filtration rate did not change during the length of the study, and SLEDAI -2K, C3, and C4 improved. The early co-management between Rheumatologists and Nephrologists is essential to prevent the rapid progression of LN. It would be interesting to administer antioxidant supplements to patients with a recent diagnosis of LN and evaluate its effect in a follow-up study. (lines 42-49)

Comment. The authors aspire to publish in a perfect journal focused on oxidative stress. In my opinion, in this article, oxidative stress is only an assessment of biomarkers - but without any more profound reflection on what results from it.

Answer. The present study aims to compare several redox-related biomarkers in patients with LN over the course of the disease. These results, accompanied by other clinical markers, may provide an overview of how redox homeostasis may influence other parameters of the disease in the long term. We expected that the modifications we have made with the aforementioned comments may strengthen the findings of this study.

Comment. I suggest the authors consider this. They should focus on the articles on oxidative stress in kidney diseases; there is no shortage of them. They should write in the article about the novelty of their results, and consider what their article brings:

Let's look:
Antioxidants 2020, 9(8), 752; https://doi.org/10.3390/antiox9080752
Antioxidants 2020, 9(11), 1079; https://doi.org/10.3390/antiox9111079
J Inflamm Res. 2023 Feb 4;16:453-465. doi: 10.2147/JIR.S399284. PMID: 36761905; PMCID: PMC9907008.
Antioxidants 2023, 12(8), 1627; https://doi.org/10.3390/antiox12081627
Lupus. 2020;29(3):311-323. doi:10.1177/0961203320904784

Answer. The Suggested references are greatly appreciated. The discussion of some of the findings was improved with information on mechanisms related to oxidative stress, SLE, and LN. The changes are highlighted in blue on lines 327-336, 366-370, 386-393, 467-478.

Comment. As before I have suggested, I also recommend inserting a flow diagram. The authors wrote: "The flow chart of the research work is shown. If you consider it should be included in the main document, we will gladly add it." I don't know where it is visible, but please include it in the publication.

Answer. The flow diagram was Added in figure 1

Comment. Finally, I think it would be helpful to have a chart/figure showing changes in parameters over time in different groups, which would show the course of changes and make it easier to follow them.

Answer. We added figure 2 with a summary of the graphic results of tables 2 and 4 corresponding to the data obtained in the 6-month follow-up.

We greatly appreciate the comments that have contributed to the improvement of the manuscript.
